

# Re-establishing glacier monitoring in Kyrgyzstan and Uzbekistan, Central Asia

Hoelzle Martin[1], Azisov Erlan[2], Barandun Martina[1], Huss Matthias[1,3],
Farinotti Daniel[3,4], Gafurov Abror[5], Hagg Wilfried[6], Kenzhebaev Ruslan[2],
Kronenberg Marlene[1,7], Machguth Horst[1,8], Merkushkin Alexandr[10],
Moldobekov Bolot[2], Petrov Maxim[9], Saks Tomas[1], Salzmann Nadine[1],
Schöne Tilo[5], Tarasov Yuri[10], Usubaliev Ryskul[2], Vorogushyn Sergiy[5],
Yakovlev Andrey[11], and Zemp Michael[8]

[1]Department of Geosciences, University of Fribourg, Fribourg, Switzerland
[2]Central Asian Institute for Applied Geosciences, CAIAG, Bishkek, Kyrgyzstan
[3]Laboratory of Hydraulics, Hydrology and Glaciology (VAW), ETH Zurich, Zurich, Switzerland
[4]Swiss Federal Institute for Forest, Snow and Landscape Research WSL, Birmensdorf, Switzerland
[5]GFZ German Research Center for Geosciences, Potsdam, Germany
[6]Department of Geography, University of Munich, Munich, Germany
[7]Meteodat GmbH, Zurich, Switzerland
[8]Department of Geography, University of Zurich, Zurich, Switzerland
[9]Glacial Geology Laboratory, Tashkent, Uzbekistan
[10]NIGMI of UzHydromet, Tashkent, Uzbekistan
[11]Uzbek scientific investigation and design survey institute, UzGIP, Tashkent, Uzbekistan

*Correspondence to:* Martin Hoelzle (martin.hoelzle@unifr.ch)

**Abstract.** Glacier mass loss is among the clearest indicators of atmospheric warming. The observation of these changes is one of the major objectives of the international climate monitoring strategy developed by the Global Climate Observing System. Long-term glacier mass balance measurements are furthermore the basis to calibrate and validate models simulating future runoff of glacierized catchments. This is essential for Central Asia, which is one of the driest continental regions of the northern hemisphere. In the highly populated regions, water shortage due to decreased glacierization potentially leads to pronounced political instability, drastic ecological changes, and endangered food security. As a consequence of the collapse of the former Soviet Union, however, many valuable glacier monitoring sites in the Tien Shan and Pamirs were abandoned. In recent years, multinational actors have re-established a set of important in-situ measuring sites to continue the invaluable long-term data series. This paper introduces the applied monitoring strategy for selected glaciers in the Kyrgyz and Uzbek Tien Shan and Pamir, highlights the existing and the new measurements on these glaciers and presents an example for how the old and new data can be combined to establish multidecadal mass balance time series. This is crucial for understanding the impact of climate change on glaciers in this region.





## 1  Introduction

Glacier fluctuations in mountain areas have been monitored in various parts of the world for more than a century (Haeberli et al., 2007; Zemp et al., 2015) and glacier changes are considered to be reliable indicators of worldwide atmospheric warming trends (IPCC, 2013). Mountain glaciers and ice caps are important for early-detection strategies in global climate-related observations. Hence, glaciers are one of the 'essential climate variables (ECVs)' in the Global Climate Observing System (GCOS). Embedded in GCOS is the Global Terrestrial Network for Glaciers (GTN-G) operated by the World Glacier Monitoring Service (WGMS), the US National Snow and Ice Data Center (NSIDC), and the Global Land Ice Measurements from Space (GLIMS) initiative. These institutions follow the so-called Global Hierarchical Observing Strategy (GHOST) (WMO, 1997b, 2010) forming the base for the strategic observation framework. The main objectives of long-term glacier monitoring are related to (1) process understanding, (2) model validation and/or calibration, (3) change detection and (4) impact assessment. Furthermore, they play a key role for assessing climate change effects such as estimating sea-level rise, regional changes in runoff, and impacts of natural hazards. Especially countries and regions that are vulnerable to climate change rely on a sound and continuous long-term database providing the necessary information to cope with future challenges in different areas such as water management, irrigation for agriculture, disaster risk reduction, and public health.

Continuous in-situ monitoring of glaciers in remote areas is a challenging task, and maintaining the necessary measurements can be both logistically and economically demanding. Reasons include missing long-term financial and/or human resources as well as general political instability, access to remote regions, natural hazards or missing infrastructure. In many countries with glacierized mountain ranges continuous observations are thus lacking. For these reasons, monitoring strategies need to be improved and different techniques such as in-situ measurements, remote sensing and modelling have to be combined to generate high-quality products.

Currently, a re-establishment of historical monitoring sites in Kyrgyzstan and Uzbekistan is jointly developed by different international projects including Capacity Building and Twinning for Climate Observing Systems (CATCOS), Central Asian Water (CAWa), CryospherIc Climate Services for improved ADAptation (CICADA) or Contribution to High Asia Runoff from Ice and Snow (CHARIS). In the frame of those projects, the measurement series on selected glaciers (see Fig. 1) in Kyrgyzstan and Uzbekistan are (re-)initiated. The project CATCOS had two phases, the first phase phase was from 2011 to 2013, which was mainly related to the technical installations and the second phase from 2014 to 2016 was mainly used for capacity building and twinning activites. The project will be continued within the next years in a cooperative effort between Kyrgyzstan, Uzbekistan, Germany and Switzerland.

This paper aims at presenting and discussing the major steps and methodologies developed and applied in Kyrgyzstan and Uzbekistan to (re-)establish glacier monitoring at the most important





sites within the two projects CATCOS and CAWa (Fig. 1). The described methods and experience might be a blue print for similar cases in other parts of the world. The key steps, around which this
paper is centered, are the (1) collection, homogenization and securing of historical data and (2) the re-establishment of glacier monitoring and (3) capacity building and twinning.

## 2   Glaciers in Central Asia

Glaciers in Central Asia constitute an important water storage component (e.g. Immerzeel et al., 2010; Kaser et al., 2010; Duethmann et al., 2015), which is of particular importance for different
sectors including agriculture and energy production (Siegfried et al., 2012). With the ongoing climate change the corresponding variations in glacier area and volume in this region are considerable. Most studies agree on the general trend of glacier mass loss, including an acceleration since the 1970s (e.g. Sorg et al., 2012; Farinotti et al., 2015). Regarding seasonal changes, the studies however disagree. In their comprehensive review on past changes in high-altitude areas of Central Asian
headwaters, Unger-Shayesteh et al. (2013) conclude that there is (i) a lack of reliable data especially for the glacio-nival zone, (ii) methodological limitation in trend analysis, and (iii) a strong heterogeneity in spatial and temporal extent of the available analyses. These restrictions hamper a sound synthesis for the whole region, and limit the understanding of interactions between changes in highly-variable climate parameters, the cryosphere, and the hydrological response of headwater
catchments. Altogether, these shortcomings indicate the importance of high-quality in-situ measurements, which are required for a better calibration and validation of local to regional-scale models for estimating future glacier mass balance and runoff.

### 2.1   Tien Shan

Glaciers in the Tien Shan cover around 12,400 km$^2$ of which about 7,400 km$^2$ are situated in Kyr-
gyzstan according to the Randolph Glacier Inventory Version 5.0 (RGIv5.0) and about 117 km$^2$ in Uzbekistan (Arendt et al., 2015). In the Tien Shan, only two continuous long-term glacier mass balance series are presently available (Tuyuksu Glacier in Kazakhstan, and Urumqihe Glacier No. 1 in China). Two long-term mass balance measurements in the Kyrgyz Tien Shan were discontinued in the 1990s (Golubin and Karabatkak). Whereas glaciers in the western part of the Tien Shan receive
winter accumulation, glaciers in the East are summer-accumulation type glaciers. Furthermore, annual precipitation sums are maximum in the Northwest and decrease south-eastwards (Voloshina, 1988). This shift goes along with an alteration of the annual precipitation maximum which is earlier in the West and in summer in the East (Dyurgerov et al., 1994). Kriegel et al. (2013) show a temperature change of 0.1-0.2 °C per decade for the ablation period (April-September) during 1960-2007
for the Naryn station. The same authors indicate an unequivocal change in precipitation across a few stations in the Central Tien Shan. A decrease in snow cover has been observed for the period from





1960 to 2007 in the entire Tien Shan (Chen et al., 2016). Precipitation changes seem to be less important. Higher air temperatures resulted in positive runoff trends in spring and autumn for the Naryn catchment, which is likely to be related to enhanced snow and glacier melt corresponding also to the

observed annual area shrinkage rates since the middle of the 20th century (e.g. Sorg et al., 2012). For Small Naryn, significant negative runoff trends were found whereas for Big Naryn a positive but not significant trend were found for August, which is the month with the largest glacier runoff contribution (Kriegel et al., 2013).

## 2.2 Pamir-Alay

In the Pamir, glaciers cover approximately $12{,}100 \, \text{km}^2$. The sub-region Pamir-Alay has a total glacier area of around $1{,}850 \, \text{km}^2$ (both values are based on RGIv5.0). In the Pamir-Alay, direct mass balance measurements only exist for Abramov Glacier (Fig. 1). For the Pamir-Alay, a strong gradient in the equilibrium line altitude (ELA) from West to East is observed (Glazirin et al., 1993). This is linked to important differences in precipitation (Suslov and Akbarov, 1973). According to Suslov and

Akbarov (1973) and Glazirin et al. (1993), the maximum annual precipitation in the Pamir-Alay is about $1900 \, \text{mm a}^{-1}$ in the western part at the border between Kyrgyzstan and Tadjikistan, and about $400 \, \text{mm a}^{-1}$ in the eastern part. The seasonal precipitation regime also differs from West to East: Whilst the West shows a maximum precipitation during autumn and winter, the East is characterized by a maximum during spring and summer. Therefore, both accumulation and ablation are region-

dependent, resulting in different glacier surface mass balance gradients. This also influences total discharge from glaciers, which is estimated as $1.6 \, \text{km}^3 \, \text{a}^{-1}$ corresponding to about 8% of the total annual runoff of all rivers in Central Asia (Suslov and Akbarov, 1973). Focusing on a region in the eastern Pamir, Khromova et al. (2006) found a reduction in glacier area of 10% from 1978 to 1990, and of 9% from 1990 to 2001. Glacier front variations with annual rates of $-11.6 \, \text{m a}^{-1}$ for very large

glaciers, $-7.3 \, \text{m a}^{-1}$ for valley glaciers and of $-3.3 \, \text{m a}^{-1}$ for smaller glaciers were observed. The high sensitivity of glaciers to summer temperatures is assumed to be responsible for the long-term retreat (Glazirin et al., 2002). The negative mass budget of the glaciers in this region indicates that increased winter and summer precipitation cannot compensate for the increase in air temperature (Khromova et al., 2006). A smaller reduction in glacier area of 3 % $\text{a}^{-1}$ for the period 2000 and

2007 was observed by Narama et al. (2010) for a region, which is close to the SE-Fergana mountain range situated in the East of the Pamir-Alay.



## 3  Instrumentation, Methods and Data

### 3.1  Monitoring strategy

The applied strategy to re-establish glacier observation networks in Central Asia is partly based on
the Tiers 2 and 3 of GHOST (WMO, 1997a, b) and the experience gained in different monitoring
projects. For glacier mass balance, the strategy is composed of several components:

1. Mass balance measurements using the glaciological method,

2. observation of the transient snow line on photographs from terrestrial automatic cameras
   and/or on satellite images during the summer months, and

3. a mass balance model driven by nearby automatic weather station data, reanalysis data, or
   climate model results (see Fig. 2).

As an important additional element, geodetic measurements should be integrated for calibration or
validation purposes depending on the given objectives. The combination of the different approaches
allows producing accurate mass balance estimates with a high temporal and spatial resolution (e.g.
Zemp et al., 2013). An important advantage of the described strategy is that all different components
can be used independently (e.g. Huss et al., 2009).

The re-establishment of the in-situ glacier monitoring has to obey certain criteria for the selection
of the glaciers. These are based on several pre-conditions such as field-site accessibility, availability
of historical data, geo-climatic distribution within mountain ranges, suitability for long-term moni-
toring based on glaciological feasibility and the availability of other measurements. Based on these
criteria, the five glaciers Abramov, Golubin, Batysh Sook, No. 354 and Barkrak Middle were selected
within the projects CATCOS and CAWa (see Fig. 1 and Tab. 1).

### 3.2  Implementation

The implementation of the measurement network within the re-establishing activities (in the follow-
ing referred to as 'new' measurement network) takes advantage of the concept described above. Our
approach is summarised below to clarify the individual steps during the re-establishment of the mass
balance series:

1. Finding appropriate collaboration partners.

2. Collection of all historical mass balance data from different sources.

3. Based on a review of these data and expert knowledge, the glacier selection was conducted
   and accordingly the new mass balance network was designed.

4. Set up of the new glaciological mass balance measurement networks on the selected glaciers
   in the years 2010 (Golubin, Batysh Sook, No. 354) , 2011 (Abramov), 2016 (Barkrak Middle).



In parallel, installation of automatic snow line cameras and new weather stations close to the glaciers in collaboration with partners in the CAWa project from Kyrgyzstan, Uzbekistan, Germany and Switzerland.

5. Selection of satellite images with optimal visibility of melt-out patterns and snow lines.

6. A mass balance model was used as an extrapolation tool to obtain from mass balance point measurements the glacier wide mass balance. Model validation was performed using snow line measurements from the automatic cameras or from satellite images. The model also served for reconstructing mass balance for periods with longer data gaps.

7. Comparison of mass balance retrieved by in situ measurements and geodetic measurements.

### 3.3 Glacier observations

#### 3.3.1 Data from WGMS and literature

Glacier monitoring in Central Asia started more than 60 years ago. During this time, several different glaciological programmes were established in the mountain ranges of Kyrgyzstan, Kazakhstan, Uzbekistan and Tajikistan. We compiled all available glacier data from the WGMS (WGMS, 2013). The total number of past glacier observations are summarized in Tab. 2, delivering basic information related to all existing measurements having an observation period longer than two years. For 17 glaciers, mass balance measurements exist with mean observation period lengths close to 20 years. Thickness change measurements for nine glaciers and repeated front variation measurements for 62 glaciers with measurement periods from 9 to 20 years are also available. Most of the mass balance measurements were initiated between 1960 and 1970 and were discontinued in the 1990s. Detailed description of stake and snow pit measurements for Abramov glacier were retrieved from Pertziger (1996).

#### 3.3.2 Glaciological measurements

The determination of the mass balance of a glacier using the so-called glaciological method is the standard method used since the earliest times of mass balance measurements (Mercanton, 1916). This robust method is widely used to determine the seasonal to annual mass change of individual glaciers. It is based on measurements of ablation at stakes, which are drilled into the ice of the ablation area, and of snow depth and density measured in snow pits in the accumulation area. The advantage of this method is the direct determination of the temporal evolution of mass balance. Calculation of the total mass change derived from point measurements, however, can be challenging since individual stake readings and snow pit data have to be extrapolated over the glacier, thus resulting in considerable uncertainties (for further details, see Østrem and Brugman, 1991; Kaser et al., 2003; Thibert et al., 2008; Cogley et al., 2011).



On the (re-)established glaciers the following measurements were carried out:

- For the ablation measurements in late summer, plastic or wooden stakes were drilled into the
  ice in the ablation area to a depth related to the expected melt rate at the corresponding altitude
  varying on the observed glaciers between around 2 to 10 m. This was done with an auger or a
  steam drill.

- In the accumulation area, several snow pits were dug to determine the annual snow accumula-
  tion and its density. Additionally, several snow depth measurements with steel or aluminium
  rods of 2 to 5 m length were performed. Theses were combined with occasional ground pen-
  etrating radar (GPR) measurements using a frequency of 800 MHz to detect snow layers of
  former years. This information allows an improved extrapolation of the highly variable snow
  distribution on the glacier (Sold et al., 2013) and reconstructing past accumulation rates back
  to almost one decade (Sold et al., 2015).

- Frontal variations in glacier length were measured at the glacier tongue using handheld GPS
  and/or were digitised based on satellite images.

The stake and snow pit locations aim at representing the former transects of measurement on the
glaciers to allow comparability. The total number of new stakes and pits, however, had to be reduced
substantially to keep the annual work load on a manageable level.

### 3.3.3 Geodetic mass balance

Geodetic mass balance measurements are useful to infer mass and volume changes of glaciers over
decadal periods (Ahlmann, 1924) providing considerable precision and high spatial resolution. Dig-
ital elevation models (DEMs) derived from different sources like topographic maps, GPS-surveys,
aerial photographs, satellite images, synthetic-aperture radar (SAR) or Light Detection and Ranging
(LiDAR) are compared to each other and the elevation differences over the glacier can be con-
verted into the glacier's mass change over a given time interval (e.g. Bauder et al., 2007; Thibert
and Vincent, 2009; Zemp et al., 2013). In general, multi-annual time periods are needed to reach an
acceptable level of accuracy (Paul et al., 2013). The most important uncertainties are due to (i) den-
sity conversion (Huss, 2013), (ii) elevation biases and errors in the co-registration of the two DEMs
(Berthier et al., 2006; Paul, 2008; Nuth and Kääb, 2011), (iii) data gaps, (Pieczonka et al., 2011;
Bolch and Buchroithner, 2008), and (iv) errors and artefacts in the DEMs.

Geodetic glacier volume changes focusing on Central Asia often rely on the Shuttle Radar Topog-
raphy Mission (SRTM) as a baseline for the year 2000 (e.g. Berthier et al., 2010; Gardelle et al., 2012,
2013; Surazakov and Aizen, 2006). However, the quality of the SRTM DEM is affected by electro-
magnetic wave penetration into the snow and ice. This can lead to major uncertainties, especially
affecting elevation changes in the accumulation area, for which a corrections have to be applied
(Berthier et al., 2006; Gardelle et al., 2012, 2013; Kääb et al., 2015) These corrections, however,





are difficult to quantify as they strongly vary with local conditions (Kääb et al., 2015). Other data sources for the geodetic method applied in Central Asia are topographic maps or stereo imagery derived from several sensors operating in the visible range of the spectrum, such as ASTER (Advanced

Spaceborne Thermal Emission and Reflection Radiometer), SPOT (Satellite pour l'observation de la Terre), Cartosat, Corona or Hexagon data (e.g. Hagg et al., 2004; Bolch et al., 2011, 2012; Pieczonka et al., 2013; Gardelle et al., 2013; Pieczonka and Bolch, 2015; Bolch, 2015; Petrakov et al., 2016). Furthermore, laser altimetry measurements, e.g. from the ICESat satellite mission operated between 2003 and 2009, offer a possibility of mass change computation for mountain glaciers (Kääb

et al., 2012; Gardner et al., 2013; Neckel et al., 2014; Farinotti et al., 2015). Elevation differences at points along the repeated tracks are interpolated and converted into volume and mass changes. Due to point measurements and uncertainties associated with interpolation, this method delivers robust results rather on large scales and not for individual glaciers.

### 3.3.4    Snow line observations

The calculation of glacier-wide annual mass balance from statistical relations using ELA and accumulation area ratio (AAR) is well established (Braithwaite and Müller, 1980; Braithwaite, 1984; Benn and Lehmkuhl, 2000; Kulkarni, 1992). It has been applied in many different mountain ranges (Kulkarni, 1992; Rabatel et al., 2008; Chinn et al., 2012; Stumm, 2011) and it is often used to relate temperature and precipitation with mass balance (Kuhn, 1984; Ohmura et al., 1992). Generally,

ELA and the annual mass balance are well correlated (Rabatel et al., 2005, 2012). The advantage of using ELA and AAR is their straightforward mapping using remote-sensing data (Rabatel et al., 2013). However, this approach relies on the calibration against long-term glaciological in-situ measurements. An interesting alternative is the use of transient snow lines. Their monitoring can be used as a good proxy for the sub-seasonal mass balance (Huss et al., 2013). The distinction between

snow-covered and snow-free zones on a glacier can be retrieved by different means such as analysis of repeated images taken by terrestrial cameras and satellites, or by mapping with GPS. This information can be used to establish and analyse snow-cover depletion curves (e.g. Parajka et al., 2012) and in combination with corresponding model approaches, the water equivalent of the winter snow cover can be directly extracted (e.g. Martinec and Rango, 1981; Schaper et al., 1999). Repeated

snow line observations during the ablation period are also used for mass balance model validation (Kenzhebaev et al., 2017; Barandun et al., 2015; Kronenberg et al., 2016).

The annual course of the snow line, and the related snow-covered area fraction (SCAF), was recently used in combination with a backward modelling approach to determine sub-seasonal mass balance values (Hulth et al., 2013; Huss et al., 2013). The approach by Huss et al. (2013) uses

transient snow line observations and meteorological information in combination with a mass balance model. Remote monitoring for glaciers with limited accessibility is thus possible. It offers also an



important backup that helps achieving a better coverage with mass balance estimates and thus to reduce future data gaps.

In the framework of the CATCOS project, we installed six terrestrial cameras (Mobotix, M15 and
M25) for snow line observation: Two cameras at Abramov glacier, two cameras at Golubin glacier, one camera at Glacier No. 354 and one camera at Barkrak Middle glacier. Every day, eight pictures are taken, transferred to a nearby CAWa meteorological station, and sent via satellite connection to a database (Schöne et al., 2013). For Glacier No. 354 (Kyrgyzstan) and Barkrak Middle glacier (Uzbekistan), data is stored locally and downloaded once a year.

### 3.3.5   Meteorological Measurements

Meteorological data constitute an important component of a complete glacier monitoring approach as a required input to mass balance models. There has been an extensive network of continuous long-term meteorological time series of air temperature and precipitation in Central Asian countries during the Soviet times (Aizen et al., 1995b). However, this network thinned out considerably during
the recent past (Unger-Shayesteh et al., 2013) .

Some new meteorological stations were installed in close vicinity of Abramov glacier in 2011 and Golubin glacier in 2013 by the CAWa project (Schöne et al., 2013). These two stations were equipped with standard meteorological and ground sensors (see detailed description in (Schöne et al., 2013)). Data access is open and facilitated by GeoForschungsZentrum (GFZ) in Potsdam and CAIAG in Kyr-
gyzstan (http://sdss.caiag.kg). Meteorological information for the glaciers Batysh Sook and Glacier No. 354 are retreived from the nearby Tien Shan meteorological station (3660 m a.s.l.). This station is operational since 1930 but was relocated in 1997. This causes a shift in the data, hampering its use for long-term studies. Data are available from Hydrometeorological Services of Kyrgyzstan. A small meteorological station was installed in the glacier forefield of Barkrak Middle glacier in 2016
by the CATCOS project.

The access to several historical meteorological time series is, for example, provided by the northern Eurasia Earth Science Partnership Initiative (NEESPI). Data sets such as Reananlysis products, e.g. NCEP/NCAR R1 (US National Centers for Environmental Prediction/US National Center for Atmospheric Research), ERA-Interim (European Centre for Medium-Range Weather Forecasts Re-
analysis) or MERRA (Modern Era Retrospective-analysis for Research and Applications) can be used to fill data gaps and to extend data records back in time (e.g. Salzmann et al., 2013; Schär et al., 2004; Schienmann et al., 2008; Schmidli et al., 2001).

### 3.3.6   Establishing multidecadal mass balance series

The combination of the methods described in the above chapters allows fully exploiting the richness
of the available data. Our approach is visualised schematically in Figure 2 and an example is shown in Figure 4. Establishing multi-decadal mass balance series for data-scarce regions or time periods





requires various data sources to be combined in an optimal way. The methodology that has been presented already in different studies (e.g. Barandun et al., 2015; Kronenberg et al., 2016; Kenzhebaev et al., 2017) is shortly summarised hereafter.

The core of the establishment of long-term mass balance series are in situ data (stake and snow pit measurements and meteorological information, such as precipitation and temperature), a mass balance model, and remote sensing products such as optical satellite sensors and terrestrial cameras. The model used can either be a simple degree-day or a more sophisticated energy balance model, which is able to produce sub-seasonal mass balance as an output (e.g. Machguth et al., 2006; Huss

et al., 2008, 2009). Depending on the objective of the application different investigations are possible with the ingredients of the strategy presented in Figure 2. In Barandun et al. (2015, for details), for example, we reanalysed mass balance data from the period 1968 – 1994 and calculated glacier-wide balances for the years without measurements to establish a continuous series covering the period 1968 – 2014 using a spatially distributed simple energy balance model (Fig. 4). The model was cal-

ibrated with seasonal mass balance data and was subsequently used to reconstruct the mass balance for the period with no measurements. Model validation was performed by using snowline observations derived from optical satellite images and, when available also from images of an automatic camera. In a final step, the resulting mass balance values covering several years or decades were compared to the geodetic glacier volume change determined based on the comparison of digital

elevation models (Barandun et al., 2015).

## 4 Investigations at individual monitoring sites

In the recent past, results of glacier monitoring activities in Central Asia (except Kazhakstan and China) are mainly based on remote sensing techniques with a focus on geodetic area and volume change assessments (see Tab. 4) or front variation measurements (Aizen et al., 2007; Konovalov

and Desinov, 2007; Niederer et al., 2007; Haritashya et al., 2009; Kääb et al., 2015; Narama et al., 2010; Hagg et al., 2013; Kriegel et al., 2013; Gardelle et al., 2013; Gardner et al., 2013; Ozmonov et al., 2013; Pieczonka et al., 2013; Khromova et al., 2014; Bolch, 2015; Pieczonka and Bolch, 2015; Farinotti et al., 2015; Petrakov et al., 2016). Between 1994 and 1998, most of the in-situ glacier observation programs were discontinued. The re-initiation of in-situ monitoring activities

on five selected glaciers, Abramov, Golubin, Batysh Sook, Glacier No. 354 and Barkrak Middle, now continues the historical measurement series and is described in the following sections with the main findings emerging from the current and historical monitoring. At some other glaciers like Karabatkak or Sary-Tor also a re-establishment programme was developed under the auspice of the Kyrgyz Institute of Water Problem and Hydropower Engineering together with the Moscow State

University within the project "Contribution to High Asia Runoff from Ice and Snow" (CHARIS) financed by USAID.


### 4.1 Abramov glacier

Abramov glacier is a valley-type glacier and is located in the northern slope of Pamir-Alay within the basin of the Vakhsh river, which is one of the largest tributary of the Amu Darya river (Fig. 1). The glacier drains into Koksu river, which has a hydrological catchment area of $58 \, \text{km}^2$ with a glacierization of roughly 51 % (Hagg et al., 2006). Abramov has a surface area of about $24 \, \text{km}^2$ (in 2013) and a volume of $2.54 \, \text{km}^3$ (Huss and Farinotti, 2012). The glacier is exposed to the North and ranges from about 3600 to 5000 m a.s.l.

Annual mean air temperature at the equilibrium line of the glacier (around 4260 m a.s.l.) is –6.5 to –8 °C (Kamnyansky, 2001). Average annual precipitation measured at 3837 m a.s.l. is about 750 mm (Glazirin et al., 1993). The glacier is assumed to have a temperate accumulation zone with cold ice near the surface in the ablation area, even though slightly negative temperatures were reported in the accumulation area by Kislov et al. (1977).

A research station at Abramov glacier was built in 1967 by the Central Asian Hydrometeorological Institute (SANIGMI) in Tashkent, guaranteeing the continuous collection of glaciological and meteorological observations. The station was destroyed in the late 1990s during politically unstable times (Fig. 3). With the station, also other valuable equipment and the measurement network was lost.

Continuous and detailed mass balance measurements exist from 1968 to 1994 (Pertziger, 1996), whereas from 1994 to 1998 seasonal mass balance values are available from WGMS (2001). Several publications have assessed the long-term mass balance of Abramov glacier. Comparing the results of these studies, differences in the order of $\pm 0.3 \, \text{m w.e. a}^{-1}$ (from 1974-1994) are revealed (Suslov and Krenke, 1980; Glazirin et al., 1993; Kamnyansky, 2001; Pertziger, 1996; WGMS, 2001; Dyurgerov, 2002; Barandun et al., 2015). Therefore, the data series were homogenised (Barandun et al., 2015). The historic mass balance measurements on Abramov glacier only show few positive years with mass balances exceeding $+0.3 \, \text{m w.e. a}^{-1}$, such as 1968/69, 1971/72, 1986/87, 1991/92 and 1992/93. Rasmussen (2013) calculated the Abramov glacier's mass balance sensitivity to air temperature change as $-0.47 \, \text{m w.e. a}^{-1}\text{C}^{-1}$ using a positive degree-day model driven with NCEP/NCAR Reanalysis data, indicating the highest sensitivity of all studied glaciers in Central Asia.

The mass balance was reconstructed according to the method describe in the method chapter and for the time period for which no direct measurements were available (Barandun et al., 2015). Such reconstructions of past mass balance time series are important to (a) fill data gaps, (b) compare past variabilities with current ones, (c) detect changes in glacier sensitivity, (d) interpret present and future impacts of glacier change. An example of such a reconstruction is shown in Fig. 4 for Abramov glacier (Barandun et al., 2015), where mass balance data was homogenised for the period 1968 – 1998 and 2012 – 2015 and reconstructed for the period 1999 – 2011 using a calibrated distributed mass balance model. The results of the reconstruction were validated using snow lines





digitised from the Landsat images. The mean mass balance for the time period 1998 – 2011 is $-0.51 \pm 0.17$ m w.e. a$^{-1}$ and it is $-0.44 \pm 0.10$ m w.e. a$^{-1}$ for 1968 – 2014 (Barandun et al., 2015).

Maximal discharge of Abramov glacier has been measured in the month of August with around 14 m$^3$ s$^{-1}$ during the first observation years in the 1960/70s (Yemelyanov, 1973). A future change in the discharge pattern based on different climate scenarios indicating an increase in annual runoff. Furthermore the seasonal peak will shift towards May to June and a decrease will likely occur in August (Hagg et al., 2007). Compared to the very high sensitivity of glacial runoff to air temperature,

precipitation changes appear to be of secondary importance (Hagg et al., 2007).

A new and fully automated meteorological station was installed in August 2011 at Abramov glacier at an altitude of 4100 m a.s.l. within a distance of about 1 km from the glacier. The station records GPS position, air temperature, relative humidity, atmospheric pressure, precipitation, wind speed and direction, shortwave incoming and outgoing radiation, as well as longwave incoming and

outgoing radiation, soil water content and soil temperatures (Schöne et al., 2013). The historical and re-established mass balance networks are shown in Figure 5. Field data were analysed by using a distributed mass balance model for extrapolating the point measurements to the entire glacier (Huss et al., 2009; Barandun et al., 2015) and the results are presented in Tab. 3. Since 2011, five field campaigns were conducted on Abramov glacier allowing the calculation of glacier-wide mass bal-

ance (see Fig. 6). Annual mass balance 2012 – 2016, i.e. after the re-establishing the monitoring programme, were negative with a mean of 0.41 m w.e. a$^{-1}$ (Tab. 3). The observed cumulative glacier frontal retreat since 1850 sums up to more than 2 km (see Tab. 5, Fig. 7).

### 4.2   Golubin glacier

Golubin glacier is situated in the Ala–Archa catchment located in the Kyrgyz Alatoo range in the

northern Tian Shan (Fig. 1). The catchment belongs to the larger Chu river basin, which drains into the Kazakh steppe. The Ala Archa catchment contains 48 glaciers. Bolch (2015) reports a total glaciated area in Ala Archa of $40.5 \pm 0.5$ km$^2$ in 1964 and $33.3 \pm 0.8$ km$^2$ in 2010. Golubin glacier covers an area of 5.5 km$^2$ based on a satellite image of 2002, has a volume based of 0.348 km$^3$ (Huss and Farinotti, 2012), and spans over an altitudinal range of 3300 to 4400 m a.s.l. The continental-

type glacier has a northern aspect in the accumulation area, and a northwestern aspect in the ablation area.

The climate in the Ala Archa region is chracterised by limited annual precipitation of around 700 mm a$^{-1}$ mainly during April to June (48%) (Aizen et al., 2006). Mean annual air temperature in the ablation area of the glacier (3450 m a.s.l.) is about 1.5 °C, calculated with a lapse rate of 0.72 °C

per 100 m from Baitik station (Aizen et al., 1995a).

Glaciological investigations on Golubin glacier started in 1958 and continued until 1994 when the monitoring programme was stopped (see Tab. 1). Between 1958 and 1973, the mass balance was predominantly positive and mainly negative afterwards (Aizen, 1988). Using imagery of several



satellites Bolch (2015) determined the geodetic mass change for the periods 1964 to 1999, and 1999
to 2010. A mass balance of $-0.46 \pm 0.24$ m w.e. a$^{-1}$ for the first period and of $-0.28 \pm 0.97$ m w.e.
a$^{-1}$ for the second period was detected. For the whole Ala Archa catchment, a glacier area change of
$-5.1\%$ from 1943 to 1977 and of $-10.6\%$ between 1977 and 2003 was observed (Aizen et al., 2006,
2007). The mass balance sensitivity to temperature change was determined by Rasmussen (2013) as
$-0.17$ m w.e. a $°$C$^{-1}$, which is substantially lower than for Abramov glacier.

In summer 2010, mass balance and length change measurements were re-initiated. In Figure 8 the
former and the new measurement network are presented. Complete information is available for six
years now. During this time, the length change measurements were performed by GPS and by anal-
ysis of satellite images. The recent observations of the glacier tongue could be connected to earlier
measurements by Aizen et al. (2006, 2007) (Fig. 7). The glaciological measurements were analysed
by using a distributed mass balance model (Huss et al., 2009) (Tab. 3, Fig. 6). The meteorologi-
cal station, installed in 2013, is situated at an altitude of 3300 m a.s.l. at a distance of 500 m away
from Golubin glacier (Schöne et al., 2013). Two further climate stations Alplager and Baitik are
located in the Ala-Archa catchment at the altitudes of 2340 m a.s.l. and 1580 m a.s.l., respectively.
The re-initiated mass balance measurements since 2010 indicate mostly negative mass changes. In
2010/2011 and 2015/16, a positive mass balance was measured (Tab. 3). The observed cumulative
glacier frontal retreat since 1861 sums up to around 1.3 km (see Table 5 and Fig. 7).

### 4.3 Batysh Sook glacier

Batysh Sook glacier (also named Suyok (Suek) Zapadniy or Glacier No. 419 in earlier studies
(WGMS, 1993; Hagg et al., 2013; Kenzhebaev et al., 2017)) is located in the Sook range in the
Central Tien Shan. The range comprises of 44 glaciers with a total area of 30.9 km$^2$ and volume
of 1.2 km$^3$ in 2007 (Hagg et al., 2013). Batysh Sook glacier (Fig. 1) belongs to the Naryn catch-
ment, major tributary of the Syr Darya river. The small glacier covered an area of around 1.2 km$^2$
in 2005 and covers an altitudinal range of 3950 to 4450 m a.s.l. with an estimated volume of around
0.033 km$^3$ $\pm 6.5\%$ in 2010 (Hagg et al., 2013). The glacier is assumed to be composed of a temperate
accumulation zone and a cold ablation area.

According to the Tien Shan meteorological station situated in a distance of around 32 km, the
mean annual temperature is $-6.0°$C. July is the month with the highest temperatures ($4.4°$C) and
January has a mean temperature of $-21.7°$C (1997 to 2014) (Kutuzov and Shahgedanova, 2009).
Mean annual precipitation is 360 mm (1997 to 2014), of which up to 75% was recorded during the
summer months (May to September).

Batysh Sook was already monitored during short periods in 1970/71, 1983/84 and from 1988 to
1991 (see Tab. 2 WGMS (1993)). Hagg et al. (2013) calculated an area loss of around 19.8% and a
corresponding volume loss of 12.1% for all glaciers within the Sook range for the time period 1956
to 2007.



Since 2010, several stakes and some snow pits are measured on an annual basis on Batysh Sook glacier. In Figure 9 the old and the new measurement networks are presented. The mass balance is analysed using a distributed mass balance model (Huss et al., 2009). The results indicate continuous negative mass balance of about –0.4 to –0.8 m w.e. a$^{-1}$ since 2011/2012 (Tab. 3). The observed cumulative glacier frontal retreat since 1975 sums up to around 0.32 km (see Tab. 5 and Fig. 7).

A reconstruction of past mass balances for the period 2004 to 2010 was performed by (Kenzhebaev et al., 2017) using a calibrated distributed temperature index mass balance model. For the reconstructed period an average annual mass balance of –0.39 ±0.26 m w.e. a$^{-1}$ was found. For the direct measured period 2011 to 2016, three different methods were applied. The first method, profile method, revealed a mass loss of –0.41 ±0.28 m w.e. a$^{-1}$, the second method, the contour line, re-
sulted in a negative mean mass balance of –0.34 ±0.20 m w.e. a$^{-1}$, and the third method, based on model extrapolation calculated a value of –0.43 ±0.16 m w.e. a$^{-1}$.

### 4.4 Glacier No. 354

Glacier No. 354 is situated in the southern part of the Akshiirak glacierized massif, Inner Tien Shan (Fig. 1). According to Aizen et al. (2006), the range contains 178 glaciers (87% of them are valley
type glaciers) covering an area of 371 km$^2$. The Akshiirak glaciers on the eastern and southern part of the mountain range drain into the Saridjaz river, a tributary of the Tarim river via Aksu river and the runoff of the glaciers on the western part contributes to the flow of the Naryn river. The glacier had a surface area of 6.41 km$^2$ in 2014 Kronenberg et al. (2016). The accumulation area is composed of three basins and the glacier tongue is directed to the Northwest. The glacier covers an
altitudinal range of around 3750 to 4650 m a.s.l. Hagg et al. (2013) performed GPR measurements on the glacier tongue and used a simplified ice mechanical approach to determine the total volume of this glacier to 0.272 km$^3$ ± 0.022 km$^3$. Pieczonka and Bolch (2015) quantified the geodetic mass balance for Glacier No. 354 (Bordu-Yushny in their study) from 1975 to 1999 as –0.79 ±0.25 m w.e. a$^{-1}$.

The Akshiirak range is closely situated to the Sook range, and the closest meteorological station is again the Tien Shan station (14 km distance). The climate is similar to the one of Bathys Sook glacier.

    Glacier No. 354 was selected to replace the previously monitored Sary-Tor glacier due to current access restrictions because of mining activities. Sary-Tor has mass balance observations based on the
glaciological method for the period of 1985 to 1989 (Dyurgerov et al., 1994) and its mass balance was reconstructed for the period 1930 to 1988 by Ushnurtsev (1991). Mass balance measurements were re-initiated in 2013 by a team of Russian and Kyrgyz scientists. The front variations of Sary-Tor indicate a retreat of 70 m from 1932 to 1943, no changes from 1943 to the mid-1970s, a retreat of 220 m from 1977 to 1995 and of 90 m from 1995 to 2003 (Aizen et al., 2007).



Kronenberg et al. (2016) presented a reconstruction of the seasonal mass balance of Glacier No. 354 from 2003 to 2010. This reconstruction is based on an analysis of the annual measurements of stakes and snow pits of the years 2010 to 2014 (Fig. 10). The glaciological data was used to calibrate a distributed mass balance model, which was then used to reconstruct the seasonal mass balances since 2003. In addition, winter accumulation measurements were performed in spring 2014 and also

used for model calibration. Furthermore, the geodetic volume change was determined based on two high-resolution satellite stereo images acquired in 2003 and 2012 and used to validate the modelled cumulative glacier-wide mass balance. For the period 2003 to 2012, an annual mass balance of $-0.40 \pm 0.10$ m w.e. $a^{-1}$ was modelled. This result corresponds well with the geodetic mass balance of $-0.48 \pm 0.07$ m w.e. $a^{-1}$ for the same period (see Tab. 3 and Tab. 4, Kronenberg et al. (2016)).

The observed cumulative glacier frontal retreat since 1972 sums up to around 0.66 km (see Tab. 5 and Fig. 7).

### 4.5   Barkrak Middle

Barkrak Middle glacier is located in the Oygaing valley in the Pskem catchment in the western Tien Shan, Uzbekistan (Fig. 1). The Pskem river drains into Chirchik river, a tributary of the Syr

Darya river. The area of the glaciers in the Pskem river basin including the Oygaing and the Upper Maydan valley (located in Kazakhstan) was 93.6 km$^2$ in the year 2007 (Semakova et al., 2016). The area of Barkrak Middle is 2.18 km$^2$ (Semakova et al., 2016) and the volume is 0.087 km$^3$ (Huss and Farinotti, 2012).

    Between 1960 and 2010, the glacierized area decreased by 23% in the Pskem river catchment and

more specifically a 16.8% decrease in glacier area was observed from 1980 to 2001 in the basin of the Oygaing river, where Barkrak Middle glacier, is located (Semakova et al., 2016). For Barkrak Middle glacier, observation of length change exist from 1971 to 1990 with a total cumulative retreat of 249 m (see Fig. 7).

    In 2016, a new mass balance network with a total of 11 stakes was established (Fig. 11). In

addition, a meteorological station in the glacier forefield and an automatic snow line camera were installed (Fig. 11).

### 5   Summary of glacier length and mass changes in Central Asia

The measurements performed on various glaciers in Central Asia show an overall mass loss over the last five decades. There are some individual years with positive mass balances within all decades

(Fig. 6, 12, Tab 4). However, the positive mass balances are mostly observed during the beginning of the measurements in the 1960/70s. Long-term trends based on the glaciological, as well as on the geodetic method indicate a mean negative mass balance in the range of $-0.3$ to $-0.8$ m w.e. $a^{-1}$. Most negative mass balances were observed at Abramov, Karabatkak, Tuyuksu and Urumqihe





WGMS (2013), whereas Golubin and Batysh Sook exhibited smaller mass loss. This observation is
in line with the mass balance sensitivities of the glaciers in Central Asia as inferred by Rasmussen
(2013). The observed cumulative mass balance determined by the glaciological method is supported
by several remote sensing studies underlining the findings (Aizen et al., 1995b; Bolch et al., 2011,
2012; Pieczonka et al., 2013; Gardelle et al., 2013; Pieczonka and Bolch, 2015; Bolch, 2015; Kro-
nenberg et al., 2016).

The network of glaciological and geodetic measurements in the Tien Shan Mountains has been
strongly improved by the re-establishment of the monitoring programmes starting in 2010. In the
Pamir Mountains, however, glacier mass balance measurements are still sparse as only Abramov
glacier in the most northwestern part is monitored. This glacier is not representative for the en-
tire Pamir, also containing glaciers at much higher altitudes that respond differently to atmospheric
warming (e.g. Kääb et al., 2015, 2012; Gardner et al., 2013; Gardelle et al., 2013).

Length changes are an important element of glacier monitoring. Direct measurements extend back
into the 19th century in several mountain ranges and some reconstructions based on geomorpholog-
ical dating go back to the 16th century (Zemp et al., 2015). In general, the cumulative front variation
measurements for the investigated glaciers in Central Asia reveal a long-term glacier retreat (see Tab.
3, Fig. 7). Using a simple parametrisation scheme developed by Haeberli and Hoelzle (1995) and
used by Hoelzle et al. (2003) to convert curves of cumulative glacier advance and retreat into time
series of temporally averaged mass balance by applying a continuity model originally proposed by
Nye (1960). This approach considers step changes after full dynamic response and new equilibrium
of the glacier based on the assumption that a mass balance change is leading to a corresponding
glacier length change depending only on the original length of the glacier and its annual mass bal-
ance (ablation) at the glacier terminus. Calculating these values for Abramov (for the period 1850 –
2016) is resulting in –0.35 m w.e. $a^{-1}$, for Golubin (1861 – 2016) around –0.2 m w.e. $a^{-1}$, for Batysh
Sook (1975 – 2016) and Glacier 354 (1972 – 2016) around to –0.35 m w.e. $a^{-1}$.

## 6 Discussion

We demonstrate that, a valuable glaciological dataset is available for the Central Asian mountains,
particularly for the Tien Shan, despite of different historical backgrounds and changing methodolog-
ical aspects. New monitoring networks in combination with historical measurements will generate
highly valuable baseline data for future research, climate change observations, assessment of climate
change impacts and related adaptation measures, including water resource management. Continuous
long-term support and cooperation among several countries and institutions is required to keep the
successfully re-established measurements alive. The newly developed strategy and its implementa-
tion to connect in-situ measurements with remote sensing and numerical modelling techniques (see
Fig. 2) will allow the establishment of an efficient glacier monitoring programme. This will guar-



antee an optimum between available human capacity and efforts, sustainable financial coverage and
data quality, offering a feasible way to perform long-term monitoring. Our approach implemented in
Kyrgyzstan and Uzbekistan has to be proven during the next years and should be further developed.
Over the last years, it delivered promising results, although there are still many improvements pos-
sible. One of the most urgent needs is capacity building, i.e. the education of young local scientists
being able to continue the monitoring programme independently (Nussbaumer et al., 2017).

Our approach also reveals that on a scientific basis, an in-depth re-evaluation of long-term geodetic
mass changes on all selected glaciers is absolutely necessary. Most geodetic measurements (e.g. for
Glacier 354, Urumqihe No. 1 and Tuyuksu) match well with the glaciological mass balances (Fig.
12). All these measurements show almost persistently negative mass balances since the mid-20th
century. At Abramov glacier, however, considerable differences between geodetic and reconstructed
glaciological measurements were found (Fig. 12 and Tab. 4). For the period 2000 to 2011, Barandun
et al. (2015) determined a negative value of $-0.51 \pm 0.17$ m w.e. $\text{a}^{-1}$, whereas Gardelle et al. (2013)
reported a mass balance of $-0.03 \pm 0.14$ m w.e. $\text{a}^{-1}$ for the same period. The geodetic measurements
are subject to considerable uncertainties that are, for example, related to underestimated penetra-
tion of the radar signals into snow and firn in the accumulation areas during the acquisition of the
SRTM mission in the year 2000 (Paul et al., 2013). This is confirmed by other studies using ICESat
measurements in the same region (Kääb et al., 2015, 2012; Gardner et al., 2013) and by ground pen-
etrating radar measurement in the accumulation area of Abramov glacier (Barandun et al., 2015).
This shows the need for further analysis of existing aerial and satellite data in order to create ad-
ditional high-accuracy elevation models. Another problem encountered during the homogenisation
process of the mass balance time series was a considerable inconsistency, which is mainly related to
different interpretations of the former stake networks or to missing measurements.

Glacier frontal variations complement the mass balance measurements. They show a continuous
retreating trend for the last decades and century. The inferred estimates of mass change calculated
based on the length change measurements are in quite good accordance with the mass balance mea-
surements.

## 7   Conclusions

This paper demonstrates the richness of historical dataset from the long-term glacier measurements
available in Central Asia and particularly in Kyrgyzstan and it introduces the (re)-established glacier
measurement network highlighting the importance of capacity building to make the efforts sustain-
able. In connection with the new glacier monitoring strategy and its current implementation, the
long-term in-situ data are fundamental to understand the relevant processes and the impact of cli-
mate change on glaciers in the Tien Shan and the Pamir-Alay and the impacts for the people living
nearby as well as in far distances downstream. Therefore, the cryospheric ECV 'glacier' is an im-





portant variable, for which sustainable long-term monitoring has to be ensured not only in industrial
countries but also in developing countries. Without the long-term measurements, questions like the
future behaviour of glaciers and their possible disappearance and the impacts for the society cannot
be answered comprehensively. Only well-validated and well-calibrated models based on sound pro-
cess understanding and high quality data will ensure projections with acceptable level of uncertainty.
These models and their outputs are the only tools for estimating future runoff evolution in one of the
driest continental regions of the northern hemisphere, where future water shortage due to decreased
glacierization has the potential to lead to pronounced political instability, drastic ecological changes
and might endanger future food security in a highly populated region. This highlights the importance
of in-situ monitoring networks of all ECVs within national and international Climate Services and
their use together with remote sensing and numerical models.

*Acknowledgements.* We thank F. Denzinger, M. Duishonakunov, M. Fischer, A. Ghirlanda, S. Gindraux, A.
Kääb, D. Kriegel, M. Kummert, N. Mölg, K. Naegeli, A. Neureiter, D. Petrakov, S. Reisenhofer, H. Rieck, J.
Schmale, A. Sharshebaev, L. Sold, P. Schuppli, A. Zubovich for their valuable support in the field. T. Bolch
is acknowledged for sharing data. The project was only possible thanks to support of the Federal Office of
Meteorology and Climatology MeteoSwiss through the project CATCOS (Capacity Building and Twinning
for Climate Observing Systems), Contract no. $7F - 08114.1$ between the Swiss Agency for Development and
Cooperation and MeteoSwiss. The CAWa (Water in Central Asia) project (www.cawa-project.net) was sup-
ported by the German Federal Foreign Office (contract no. AA7090002) as a part of the "Berlin Process". This
study was also supported by the Swiss National Science Foundation (SNSF), grant $200021_155903$ and the The
European Research Council (FP/2007-2013, ERC grant agreement no. 320816). We are also grateful to all col-
laborators of the Central Asian Institute for Applied Geosciences for their continued support of this long-term
project.



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



**Table 1.** Information of observed glaciers and type of measurements currently available from WGMS (2013) for mass balance (mb) and frontal variation, see Fig. 1 for location

| Glacier | Type | Period | Interval | | Re-start |
|---|---|---|---|---|---|
| Golubin | mb | 32 | 1968 | 2016 | 2010 |
| Golubin | fv | 30 | 1861 | 2016 | 2011 |
| Abramov | mb | 36 | 1967 | 2016 | 2011 |
| Abramov | fv | 17 | 1850 | 2016 | 2011 |
| Batysh Sook | mb | 11 | 1970 | 2016 | 2010 |
| Batysh Sook | fv | 17 | 1975 | 2016 | 2010 |
| Glacier No. 354 | mb | 6 | 2010 | 2016 | 2010 |
| Glacier No. 354 | fv | 14 | 1972 | 2016 | 2010 |
| Barkrak Middle | mb | - | - | - | 2016 |
| Barkrak Middle | fv | 18 | 1970 | 1990 | 2016 |

**Table 2.** Information on glaciers before the re-establishment of the measurements was initiated based on WGMS (2013). Abbreviations: KG = Kyrgyzstan, KZ = Kazakstan, UZ = Uzbekistan, TJ = Tajikistan, mb = mass balance measurements, tc = geodetic thickness change, fv = front variations

| Country | Meas. | Glaciers observed | Mean observation period | mean start date | mean end date |
|---|---|---|---|---|---|
| KG | mb | 7 | 19 | 1973 | 1994 |
| KG | tc | 2 | 2 | 1977 | 1989 |
| KG | fv | 19 | 9 | 1954 | 2002 |
| KZ | mb | 9 | 19 | 1974 | 1992 |
| KZ | tc | 7 | 2 | 1958 | 1998 |
| KZ | fv | 11 | 16 | 1957 | 1987 |
| UZ | mb | 0 | | | |
| UZ | tc | 0 | | | |
| UZ | fv | 11 | 20 | 1965 | 1988 |
| TJ | mb | 1 | 3 | 1983 | 1985 |
| TJ | tc | 0 | | | |
| TJ | fv | 21 | 11 | 1968 | 1989 |





**Table 3.** Glaciological mass balance for Abramov, Golubin, Batysh Sook and Glacier No. 354 after the re-establishment of the measurements, calculated for the hydrological year using a mass balance model

| glacier | year | area | ELA | $B_a$ |
|---|---|---|---|---|
| | | (km$^2$) | (m a.s.l.) | (m w.e. a$^{-1}$) |
| Abramov | 2011/12 | 24.06 | 4265 | $-0.795 \pm 0.30$ |
| Abramov | 2012/13 | 24.01 | 4225 | $-0.436 \pm 0.34$ |
| Abramov | 2013/14 | 24.01 | 4245 | $-0.730 \pm 0.10$ |
| Abramov | 2014/15 | 23.94 | 4115 | $+0.171 \pm 0.30$ |
| Abramov | 2015/16 | 23.93 | 4185 | $-0.274 \pm 0.30$ |
| Golubin | 2010/11 | 5.47 | 3775 | $+0.029 \pm 0.2$ |
| Golubin | 2011/12 | 5.47 | 3875 | $-0.318 \pm 0.2$ |
| Golubin | 2012/13 | 5.45 | 3835 | $-0.251 \pm 0.2$ |
| Golubin | 2013/14 | 5.45 | 4325 | $-0.665 \pm 0.2$ |
| Golubin | 2014/15 | 5.44 | 4315 | $-0.565 \pm 0.2$ |
| Golubin | 2015/16 | 5.44 | 3745 | $+0.130 \pm 0.2$ |
| Batysh Sook | 2010/11 | 1.13 | 4255 | $-0.375 \pm 0.17$ |
| Batysh Sook | 2011/12 | 1.13 | 4265 | $-0.476 \pm 0.15$ |
| Batysh Sook | 2012/13 | 1.12 | 4255 | $-0.368 \pm 0.16$ |
| Batysh Sook | 2013/14 | 1.11 | 4265 | $-0.463 \pm 0.16$ |
| Batysh Sook | 2014/15 | 1.11 | 4305 | $-0.822 \pm 0.15$ |
| Batysh Sook | 2015/16 | 1.11 | 4265 | $-0.424 \pm 0.14$ |
| Glacier 354 | 2010/11 | 6.47 | 4195 | $-0.41 \pm 0.24$ |
| Glacier 354 | 2011/12 | 6.44 | 4205 | $-0.46 \pm 0.26$ |
| Glacier 354 | 2012/13 | 6.42 | 4225 | $-0.55 \pm 0.25$ |
| Glacier 354 | 2013/14 | 6.41 | 4275 | $-0.64 \pm 0.22$ |
| Glacier 354 | 2014/15 | 6.41 | 4365 | $-0.84 \pm 0.24$ |
| Glacier 354 | 2015/16 | 6.40 | 4155 | $-0.23 \pm 0.24$ |

**Table 4.** Observed geodetic mass balance for Abramov, Golubin, Batysh Sook and Glacier No. 354

| glacier | method | source | period | $B_a$ |
|---|---|---|---|---|
| | | | | (m w.e. a$^{-1}$) |
| Abramov | $B_{glac.-rec.}$ | (Barandun et al., 2015) | 2000−2011 | $-0.51 \pm 0.15$ |
| Abramov | $B_{geod.}$ | (Gardelle et al., 2013) | 2000−2011 | $-0.03 \pm 0.14$ |
| Golubin | $B_{geod.}$ | (Bolch, 2015) | 1964−1999 | $-0.46 \pm 0.24$ |
| Golubin | $B_{geod.}$ | (Bolch, 2015) | 1999−2012 | $-0.28 \pm 0.97$ |
| No. 354 | $B_{geod.}$ | (Pieczonka and Bolch, 2015) | 1975−1999 | $-0.79 \pm 0.25$ |
| No. 354 | $B_{geod.}$ | (Kronenberg et al., 2016) | 2003−2012 | $-0.48 \pm 0.07$ |





**Table 5.** Length change $\Delta L$ measurements for Abramov, Golubin, Batysh Sook and Glacier 354 based on GPS surveys, satellite images and maps

| Abramov | | Golubin | | Batysh Sook | | Glacier 354 | |
|---|---|---|---|---|---|---|---|
| Period | $\Delta L$ (m) | Period | $\Delta L$ (m) | Period | $\Delta L$ (m) | Period | $\Delta L$ (m) |
| 1850-1900 | −572 | 1861-1883 | −131 | 1975-1977 | −25 | 1972-1977 | −89 |
| 1900-1936 | −232 | 1883-1905 | −100 | 1977-1994 | −47 | 1977-1998 | −148 |
| 1936-1954 | −169 | 1905-1927 | −110 | 1994-1998 | −24 | 1998-1999 | −47 |
| 1954-1964 | −125 | 1927-1949 | −125 | 1998-1999 | −22 | 1999-2001 | −35 |
| 1964-1973 | −70 | 1949-1955 | −72 | 1999-2001 | −27 | 2001-2002 | −39 |
| 1973-1980 | −92 | 1955-1962 | −42 | 2001-2002 | −18 | 2002-2006 | −33 |
| 1980-1986 | −52 | 1962-1967 | −22 | 2002-2006 | −27 | 2006-2007 | −23 |
| 1986-1992 | −193 | 1967-1972 | −17 | 2006-2007 | −15 | 2007-2009 | −27 |
| 1992-2000 | −201 | 1972-1975 | −47 | 2007-2008 | −14 | 2009-2010 | −18 |
| 2000-2007 | −102 | 1975-1976 | −26 | 2008-2009 | −18 | 2010-2011 | −25 |
| 2007-2008 | −27 | 1976-1977 | −6 | 2009-2010 | −23 | 2011-2013 | −36 |
| 2008-2009 | −39 | 1977-1978 | −30 | 2010-2011 | −20 | 2013-2014 | −22 |
| 2009-2010 | −69 | 1978-1989 | −39 | 2011-2012 | −15 | 2014-2015 | −45 |
| 2010-2012 | −22 | 1989-1992 | −65 | 2012-2013 | −6 | 2015-2016 | −76 |
| 2012-2013 | −32 | 1992-1993 | −7 | 2013-2014 | −9 | | |
| 2013-2014 | −27 | 1993-1994 | −8 | 2014-2015 | −9 | | |
| 2014-2016 | −38 | 1994-1995 | −7 | 2015-2016 | −3 | | |
| | | 1995-1996 | −20 | | | | |
| | | 1996-1997 | −16 | | | | |
| | | 1997-2000 | −71 | | | | |
| | | 2000-2001 | −34 | | | | |
| | | 2001-2002 | −37 | | | | |
| | | 2002-2006 | −79 | | | | |
| | | 2006-2008 | −27 | | | | |
| | | 2008-2011 | −22 | | | | |
| | | 2011-2012 | −12 | | | | |
| | | 2012-2013 | −38 | | | | |
| | | 2013-2014 | −32 | | | | |
| | | 2014-2015 | −25 | | | | |
| | | 2015-2016 | −27 | | | | |





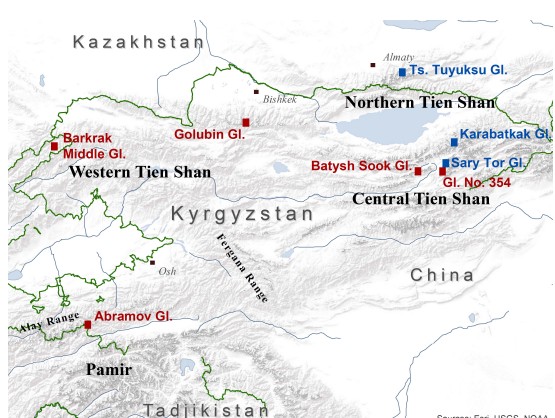

**Figure 1.** Map of glaciers in Central Asia, where the investigations were gradually re-established starting in 2010. Red symbols show glaciers investigated within the CATCOS/CAWa projects and blue glaciers are covered by other projects such as CHARIS

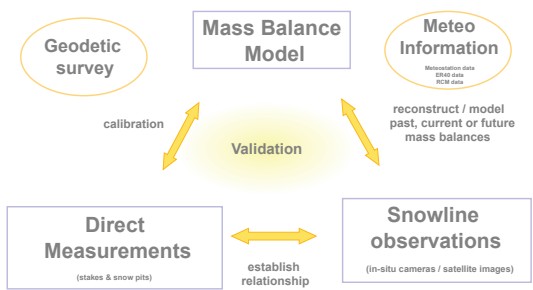

**Figure 2.** Schematic view of the monitoring strategy applied for re-established measurements in Central Asia

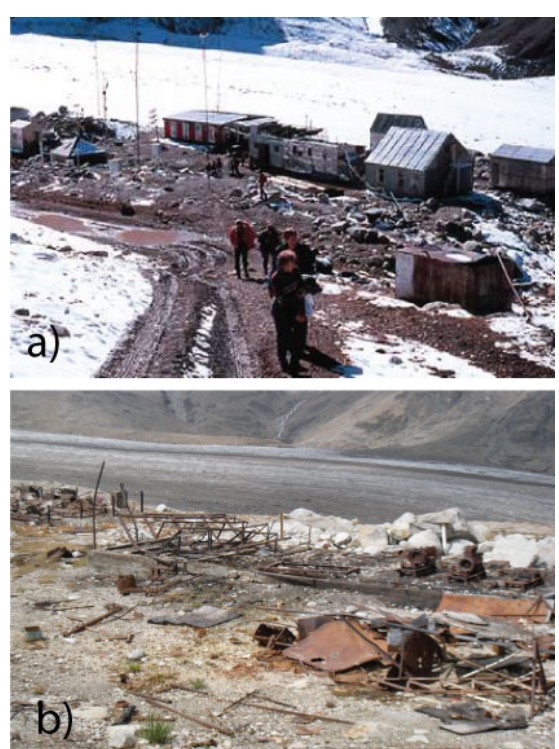

**Figure 3.** (a) Abramov research station in the year 1993 (photo: NSIDC), (b) Ruins of the Abramov research
station in 2011 (photo: M. Hoelzle)

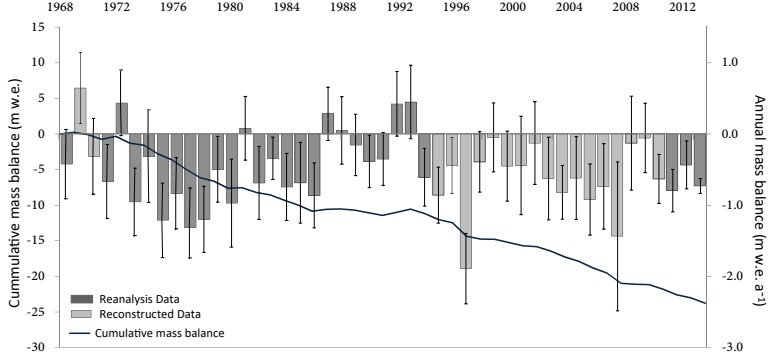

**Figure 4.** Reconstructed mass balance data for Abramov glacier according to Barandun et al. (2015)





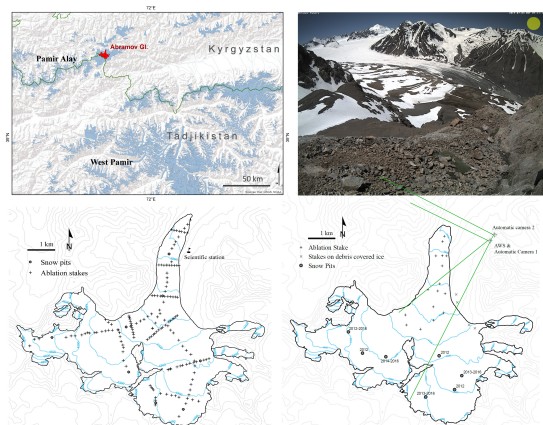

**Figure 5.** Historical and new mass balance network on Abramov glacier with corresponding map (upper left) and snow line camera picture (upper right)

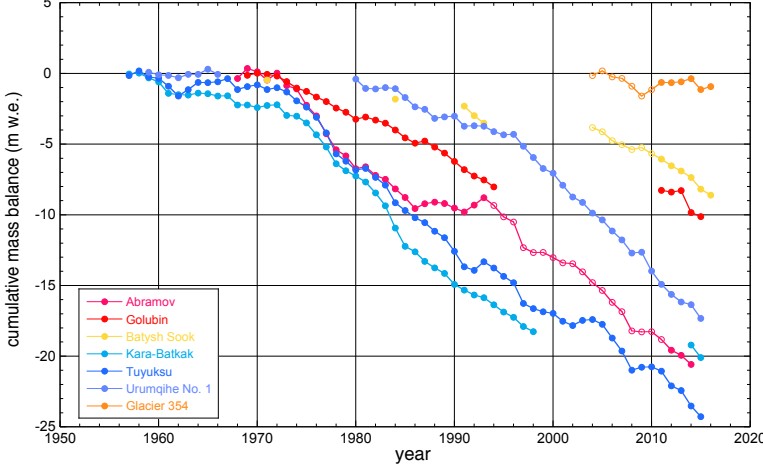

**Figure 6.** Cumulative mass balance measurements for glaciers in Central Asia. Filled dots are direct glaciological measurements, non-filled dots are reconstructed values.





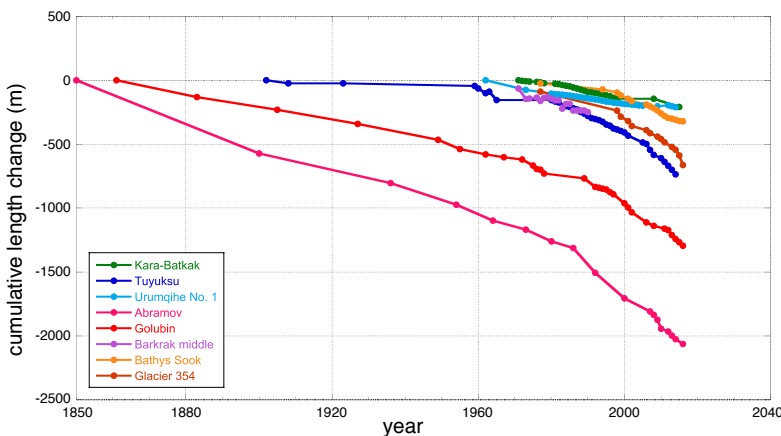

**Figure 7.** Cumulative front variation measurements for glaciers in Central Asia

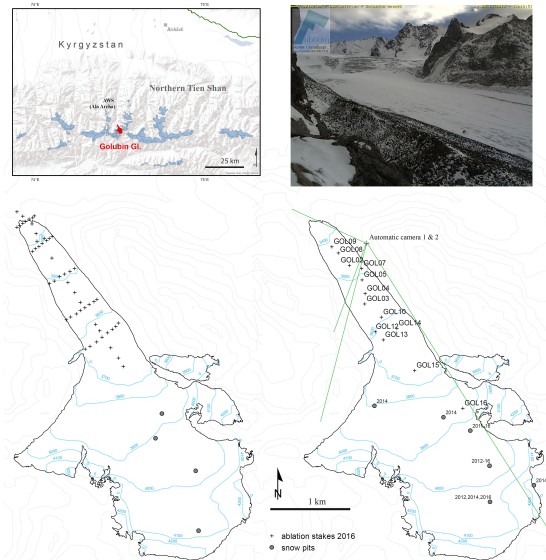

**Figure 8.** Historical and new mass balance network on Golubin glacier with corresponding map (upper left) and snow line camera picture (upper right)

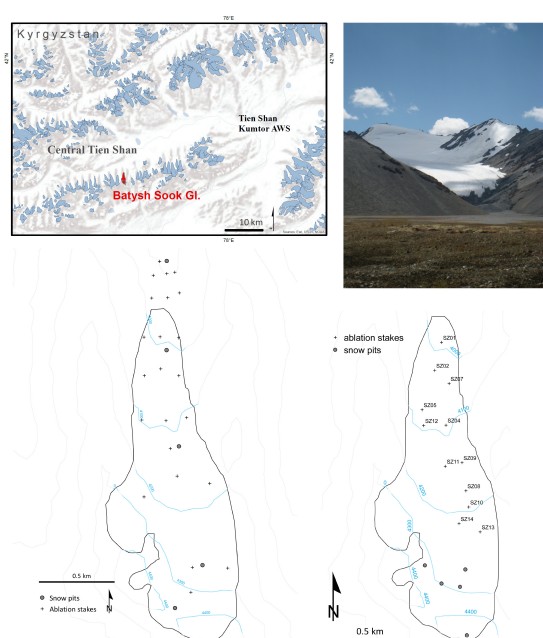

**Figure 9.** Historical and new mass balance network on Batysh Sook glacier with corresponding map (upper

left) and picture (upper right)





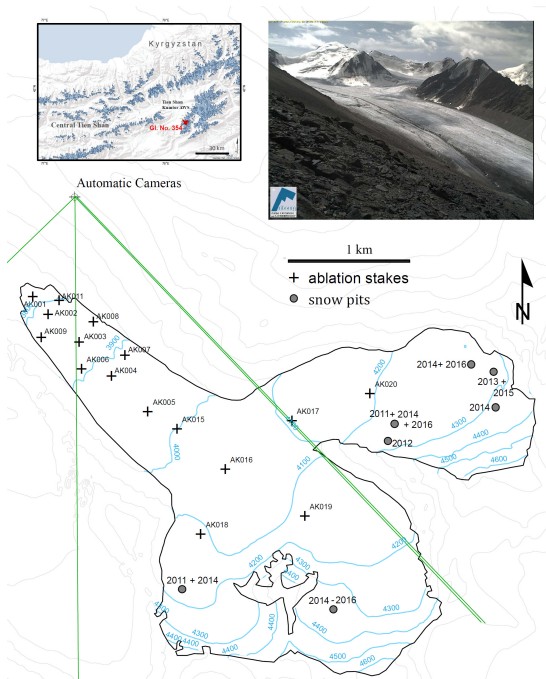

**Figure 10.** Mass balance network on Glacier No. 354 with corresponding map (upper left) and snow line camera (upper right)

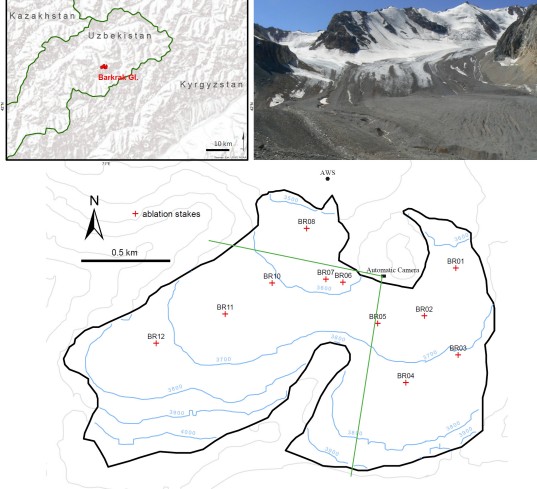

**Figure 11.** Mass balance network on Barkrak Middle with corresponding map (upper left) and picture (upper right)



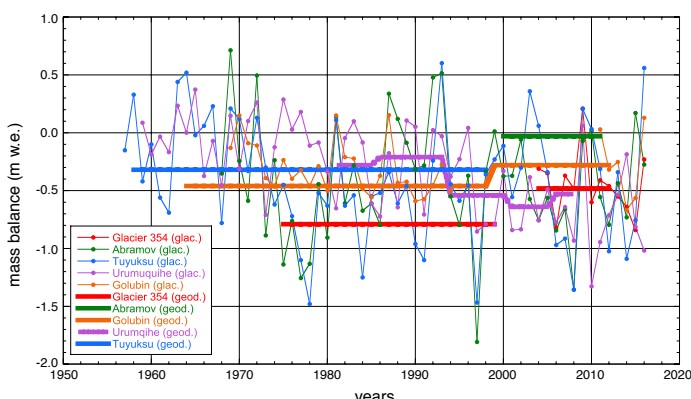

**Figure 12.** Comparison of glaciological and geodetic mass balance for glaciers in Central Asia. The geodetic mass balances are from (Hagg et al., 2004) (Tuyuksu), (Wang et al., 2014) (Urumqihe no. 1), (Pieczonka and Bolch, 2015; Kronenberg et al., 2016) (Glacier 354), and (Gardelle et al., 2013) (Abramov). The glaciological data are provided by WGMS (2013).