# Peer review of "Re-establishing glacier monitoring in Kyrgyzstan and Uzbekistan, Central Asia"

_Geoscientific Instrumentation, Methods and Data Systems, 2017_

## Referee Comment (RC1) · Anonymous Referee #1 · 6 Jun 2017

General comments. In this paper the authors describe attempts to reestablish long-term mass balance monitoring programs at Central Asian glaciers. They also review and analyses the former data from glaciers in the region and summarize the results. The paper is well written, the structure is fine and the language is clear. This is a very useful paper both as a review of former data and as for the methodologies to restart the series. The discussion and conclusions are sound. They present data from a region that is lacking data and thus it is important that these old data is taken care of and likewise that new monitoring programs are initiated.

Some more detailed comments. The referencing is appropriate and since a large part of the paper is a review of older data, probably the large number of references is needed, but more than seven pages with more than one hundred references is a lot

for a fairly short paper. Section 7 Conclusions contains arguments for why it is important to maintain these series rather than a summary of the results from the scientific data. These results are given in section 5. However, I think this structure is fine in a paper like this where the main aim is towards justifying the reestablishment of the long time-series. In the title of the paper they say glacier monitoring in Kyrgyzstan and Uzbekistan. However, in the text they do not say which glacier are in which country and in Fig. 1 Uzbekistan does not appear but all the five glaciers they discuss are given on the map. The map in Fig.1 could be improved. The boarders between the countries seem to be the green line, but this line is not continuous. In section 2.2 the heading is Pamir-Alay, but this name does not appear in the map. Uzbekistan does appear in fig. 11. In Fig. 6 it is impossible in my print-out to distinguish between filled and non-filled dots. I can see it if I enlarge the pdf so maybe this is not a problem in open-access. The same is the case for the maps in Figs. 5, 8, 9 and 10. It is very hard to read any numbers in a print-out but enlarged to 200% on the screen it is OK.

---

## Referee Comment (RC2) · Anonymous Referee #2 · 20 Jun 2017

General comments.

In this paper the authors introduce a monitoring strategy for selected glaciers in the Kyrgyz and Uzbek Tien Shan and Pamir, highlights the existing and the new measurements on these glaciers. The paper is well written and clear and demonstrates the richness of historical database. The references are appropriate and in fact a large part of the paper is a review of exixting data. Although, maybe it seems too much.

Probably the authors have to include how the old and new data can be combined together to establish multidecadal mass balance time series. This is crucial for understanding "the example" that they are trying to present . It would highlight the importance of in-situ monitoring networks of all ECVs within national and international Climate Services and their use together with remote sensing and numerical models.

[Figure]

Some maps (such as Fig.1 or5, 6 or 8) have to be improved in order to differentiate lines and dots.

---

## Author Comment (AC1) · 15 Jul 2017

**Coverletter**

**Re-establishing glacier monitoring**
**in Kyrgyzstan and Uzbekistan, Central Asia**
**Hoelzle et al.**

Dear Editor,

We would like to thank you and the two anonymous reviewers for their constructive remarks and comments on our paper. We have revised the paper, as well as the figures and tables according to the comments of the two reviewers. You will find our corresponding answers and changes in this cover letter indicating our responses (normal font style) to the reviewers' comments (displayed in italic font style).

We would like to thank again for your efforts.

On behalf of all co-authors

Martin Hoelzle

**Comments of Reviewer #1**

*Comment 1:*
*The referencing is appropriate and since a large part of the paper is a review of older data, probably the large number of references is needed, but more than seven pages with more than one hundred references is a lot for a fairly short paper.*

Answer 1:
Thanks for this comment. As it is stated by reviewer#1 our paper partly represents a review of older material and previous research in this region. We are therefore obliged to acknowledge the published literature on which parts of our study is based. We consider such a complete compilation of previous studies in the field of glacier monitoring in Central Asia are important and valuable and, hence, would like to keep the references as they are.

*Comment 2:*
*Section 7 Conclusions contains arguments for why it is important to maintain these series rather than a summary of the results from the scientific data. These results are given in section 5. However, I think this structure is fine in a paper like this where the main aim is towards justifying the reestablishment of the long time-series.*

Answer 2:
We agree with reviewer#1 that our conclusions are not always directly related to our results. However, our paper is not a 'standard' research paper but rather represents a review that links older studies to a new monitoring strategy. We thus present previous research finding as well as the results of the new monitoring efforts. A conclusion focusing on the quantitative results of a few years of recent monitoring would not be appropriate in this context. We think it is justified to keep the general aspects related to the monitoring in the conclusion section as it is one of our main goals of the paper to show how long-term monitoring series can be linked with new measurement approaches and how abandoned time series can be re-established and consistently be continued.

*Comment 3:*
*In the title of the paper they say glacier monitoring in Kyrgyzstan and Uzbekistan. However, in the text they do not say which glacier are in which country and in Fig. 1 Uzbekistan does not appear but all the five glaciers they discuss are given on the map. The map in Fig.1 could be improved. The boarders between the countries seem to be the green line, but this line is not continuous.*

Answer 3:
We have corrected the figures and adjusted the manuscript text according to the suggestions of reviewer#1. We have now added the starting date of the monitoring for each glacier and better pointed out its location. In addition, we have reworked Fig. 1 showing Uzbekistan's boundaries clearer. We added the new figure to our response.

*Comment 4:*
*In section 2.2 the heading is Pamir-Alay, but this name does not appear in the map.*

Answer 4:
Figure 1 is adjusted according to the reviewers comment and 'Pamir-Alay' is shown in the map. We added the new figure to our response.

*Comment 5:*
*Uzbekistan does appear in fig. 1.*

Answer 5:
Figure 1 is adjusted. We added the new figure to our response.

*Comment 6:*
*In Fig. 6 it is impossible in my print-out to distinguish between filled and non-filled dots.*

Answer 6:
We clarified the distinction between filled and non-filled dots in Figure 6. We added the new figure to our response.

*Comment 7:*
*The same is the case for the maps in Figs. 5, 8, 9 and 10.*

Answer 7:
We reworked figure 5, 8, 9 and 10 according to the reviewer's suggestions. We added this figures to our response.
* * *
Comments of Reviewer #2

*Comment 1:*
*Probably the authors have to include how the old and new data can be combined together to establish multi-decadal mass balance time series. This is crucial for understanding "the example" that they are trying to present. It would highlight the importance of in-situ monitoring networks of all ECVs within national and international Climate Service and their use together with remote sensing and numerical models.*

Answer 1:
Thanks for this important comment. As we have already published several papers in this region focusing exactly on the approach for establishing multi-decadal mass balance time series (Barandun et al. (2015), Kronenberg et al. (2016), Kenzhebaev et al. 2017) we wish to prevent a repetition of describing the methods used in the published articles again in detail. Therefore, we present a short summary and instead make reference to those articles in chapter '3.3.6 Establishing multi-decadal mass balance series'.

*Comment 2:*
*Some maps (such as Fig.1 or 5, 6 or 8) have to be improved in order to differentiate lines and dots.*

Answer 2:
We reworked the corresponding figures. Please see our answers to the same remarks of reviewer#1 in comments 6 & 7.